# Dietary Lignans: Definition, Description and Research Trends in Databases Development

**DOI:** 10.3390/molecules23123251

**Published:** 2018-12-08

**Authors:** Alessandra Durazzo, Massimo Lucarini, Emanuela Camilli, Stefania Marconi, Paolo Gabrielli, Silvia Lisciani, Loretta Gambelli, Altero Aguzzi, Ettore Novellino, Antonello Santini, Aida Turrini, Luisa Marletta

**Affiliations:** 1CREA Research Centre for Food and Nutrition, Via Ardeatina 546, 00178 Rome, Italy; massimo.lucarini@crea.gov.it (M.L.); emanuela.camilli@crea.gov.it (E.C.); stefania.marconi@crea.gov.it (S.M.); paolo.gabrielli@crea.gov.it (P.G.); silvia.lisciani@crea.gov.it (S.L.); loretta.gambelli@crea.gov.it (L.G.); altero.aguzzi@crea.gov.it (A.A.); aida.turrini@crea.gov.it (A.T.); luisa.marletta@crea.gov.it (L.M.); 2Department of Pharmacy, University of Napoli Federico II, Via D. Montesano 49, 80131 Napoli, Italy; ettore.novellino@unina.it (E.N.); asantini@unina.it (A.S.)

**Keywords:** dietary lignans, national databases, food groups, dietary intake, harmonized databases

## Abstract

The study aims to communicate the current status regarding the development and management of the databases on dietary lignans; within the phytochemicals, the class of the lignan compounds is of increasing interest because of their potential beneficial properties, i.e., anticancerogenic, antioxidant, estrogenic, and antiestrogenic activities. Furthermore, an introductory overview of the main characteristics of the lignans is described here. In addition to the importance of the general databases, the role and function of a food composition database is explained. The occurrence of lignans in food groups is described; the initial construction of the first lignan databases and their inclusion in harmonized databases at national and/or European level is presented. In this context, some examples of utilization of specific databases to evaluate the intake of lignans are reported and described.

## 1. Introduction

Within phytochemicals, phenolic compounds called lignans have attracted the interest of food chemists and nutrition researchers over the years. Lignans are vascular plant secondary metabolites, with widespread occurrence in the plant kingdom, and which are ascribed a wide range of physiological functions, positively affecting human health [1]. They are a class of secondary plant metabolites that belong to the group of diphenolic compounds derived from the combination of two phenylpropanoid C6–C3 units at the β and β’ carbon, and can be linked to additional ether, lactone, or carbon bonds; they have a chemical structure like the 1,4-diarylbutan [2]. The range of their structures and biological activities is broad. They are derived from the shikimic acid biosynthetic pathway [3]. The range relative to structurally different forms of lignans and biological activities is broad [4,5]. The main commonly studied and reported compounds are secoisolariciresinol, lariciresinol, matairesinol, pinoresinol, medioresinol, and syringaresinol (shown in Figure 1), while, recently, the isolation and structure elucidation of new lignan compounds have been carried out [6,7,8] and the spectrum of their attributing properties has been widened [9,10,11].

Plant lignans give rise to metabolites, enterodiol, and enterolactone [12], generally called enterolignans due to their colonic origin (named also mammalian lignans) (shown in Figure 2).

Enterolignans, and some of their plant precursors, are reported to have several biological activities—antitumorigenic [13], anticarcinogenic [14], estrogenic or anti-estrogenic [15,16], as well as antioxidant properties [17].

Lignans, in line with other natural compounds, contribute in disease prevention and health promotion [18,19]; several studies have showed the potential of lignan-rich diets against the development of various diseases, particularly hormone-dependent cancer, cardiovascular diseases, and diabetes [20,21,22,23,24,25,26,27]. 

Lignans are the basis for novel perspectives for health promotion and disease prevention as nutraceuticals and functional foods [28,29,30,31,32]. Currently, Pilkington, [33], by using a chemometric approach, have analyzed the physicochemical properties of classical lignans, neolignans, flavonolignans, and carbohydrate–lignan conjugates to assess their absorption, distribution, metabolism, excretion and toxic (ADMET) profiles, and establish if these compounds are lead-like/drug-like and, thus, have potential to be, or act as, a driver in the development of future therapeutics; the results showed how carbohydrate–lignan conjugates and flavonolignans are less drug-like, while lignans showed a particularly high level of drug-likeness [33]. 

Nowadays, lignan species and their quantity in food products are determined. Different methodologies have been defined for the extraction and identification of lignans [34,35,36,37,38,39,40]. The extraction procedure from the food matrix represents a key issue and, in particular, the type of hydrolysis step (alkaline, acid hydrolysis, enzymatic hydrolysis, or a mixture of them). The expanding demand for lignans are stimulating the interest in identification of new sources and in improvement of analytical and purification procedures. Analytical values using HPLC, as well as either gas or liquid chromatography–mass spectrometry, were developed and carried out [41,42]. The development and the assessment of methodologies for the extraction, identification, and determination of lignans are achieved [17,43,44]. Also, the “new” emerging lignans, due to LC combined with HR-MS/MS, have been, and will continue, broadening the view regarding dietary lignans [45]; simultaneously, the synthesis [46,47] and the design [48] of new compounds are being carried out. 

The complex relationship between food, nutrition, and health [49] is explored via nutrients and bioactive compounds, i.e., beneficial food components [50], and via non-beneficial food components [51]. In this direction, a directory of information about bioactive component databases, specialized, at a national and European level, is being developed, and will be useful for the planning and evaluation of clinical and epidemiological research studies on bioactive components. Databases of lignans in food products are being creating in several countries (Finland, Netherlands, United States, Canada, United Kingdom, Japan, and Spain), and represent the first step for establishing comprehensive and harmonized dietary databases, including all or nearly all bioactive compounds [1]. Reliable methods of exposure measurement are essential for understanding the potential benefits of lignans [52].

## 2. Databases: Significance, Principles and Common Criteria/Measures

Databases, also called electronic databases, represent a system to generate and collect any data, information, and documentation specially organized for rapid search and retrieval by a computer [53]. Databases are tools constructed to facilitate the storage, retrieval, modification, and deletion of data in conjunction with various data-processing operations [54].

A comprehensive food composition database (FCDB) should be a repository of all numeric, descriptive, and graphical information on the nutrient characteristics of foods [55]; the term food composition data indicates all information referring to the description and identification of foods and their food components (nutrient values, number of sample collections and analyses, analytical methods, descriptive coding, photos, data source, value documentation, etc.) and include various steps in the production, generation, compilation, and publication of data [55].

The EuroFIR project (European Food Information Resource Network of Excellence) was born to develop and integrate a comprehensive, coherent, and validated network of databanks providing a single, authoritative source of food composition data for Europe [56,57]. In this project, efforts in developing procedures for defining and establishing a standardized approach of study have been carried out from the various European partners within their FCDB [56,57].

The establishment of the “Project Committee—Food composition data” (CEN/TC 387, 2008–2013) [58] was an important milestone for the EuroFIR Network of Excellence to reach this objective. A common European standard, established within the CEN-European Committee for Standardization framework, represents a key tool enabling unambiguous identification and description of food composition data and its quality in e.g. databases, for dissemination and interchange [58]. 

Generally, the use of database management system allows the administration of large volumes of information and data by providing epidemiological research to store large varieties of food consumed for each individual subject and the comparability of data, representing a basic tool for obtaining reliable information on the relationship between nutrients and foods [59,60].

The utilization made by different users requires that FCDBs follow very specific compilation criteria, such as representativeness, accuracy in the production and selection of analytical values, traceability of data taken from other sources at the nutrient level, and clarity in the designation and description of the food [60].

In this context, the food grouping systems in food composition databases represent a key tool. Currently, Durazzo et al. [60] summarized and discussed how the food grouping systems of the various international food composition databases (FCDBs), in terms of number, type and class of consumed foods (e.g., ingredients, commercial products, cooked food, recipes, mixed dishes, etc.) vary between different countries (usually, 10 and 25 food groups), and are constantly evolving according to their changes and updates; the authors marked how these groupings are structured according to the convenience of using the nutritional composition of specific foods and, therefore, there is not an internationally standardized approach.

## 3. Distribution of Lignans in Food: Occurrence

Lignans are in a wide variety of plants from different origins, including the major edible plants. Amongst the latter, flaxseed and sesame seeds represent rich sources of lignans [40,61,62,63,64,65], whereas wood knots in coniferous trees, particularly Norway spruce, are identified as the most concentrated lignan sources known so far [66]. 

The main sources of dietary lignans are oilseeds (i.e., flax, soy, rapeseed, and sesame), whole-grain cereals (i.e., wheat, oats, rye, and barley), legumes, various vegetables and fruit (particularly berries), as well as beverages, such as coffee, tea, and wine, and, recently, lignans are also reported in dairy products, meat, and fish [64,65,67,68,69,70,71,72,73,74,75,76,77,78,79,80,81,82,83,84]. The types and amounts vary from one source to another. The content of some lignans, as well as the degree of esterification of their glycosides, could vary with different growing conditions, geographic location, climate, and genetic characteristics. Some examples of profile and distribution of lignans in common food groups are here reported, from research in the literature applying different methodological approaches. As concluded by Durazzo et al. [17], in a systematized assessment of lignans in cereals and cereal-based products for grains studied in [65,73,76], the total average values in grains ranged between 23 and 401 µg/100 g dry weight, with lariciresinol the most representative. As, for instance, for vegetables, Milder et al. [64] reported a content of total lignans (as the sum of secoisolariciresinol, matairesinol, lariciresinol, and pinoresinol, and expressed as µg/100 g fresh edible weight) of 1325 for broccoli, 185 for cauliflower, 787 for white cabbage, 171 for carrot, 58 for tomato, and 48 for chicory. Another example was given by Penalvo et al. [65] that described, for asparagus, a following profile of lignan concentrations: secoisolariciresinol 183 µg/100 g wet basis, syringaresinol 58 µg/100 g wet basis, pinoresinol 49 µg/100 g wet basis, lariciresinol 47 µg/100 g wet basis, medioresinol 5 µg/100 g wet basis, matairesinol 2 µg/100 g wet basis whereas, for eggplant, tomato, and radish, the most representative was lariciresinol [65]. For the fruit group, as reported in a work of Kuhnle [75] secoisolariciresinol and matairesinol were identified, respectively, in orange (peel and pith removed, 21 and <1 µg/100 g wet weight), nectarine (stoned, 24 and <1 µg/100 g wet weight), apricot (stoned, 51 and <1 µg/100 g wet weight), mango (skinned and stoned, 17 and 1 µg/100 g wet weight), melon (cantaloupe, skin and seeds removed, 16 and <1 µg/100 g wet weight), and others [75]. Moreover, Penalvo et al. [70] showed for avocado, a profile of decreasing concentration of lignans, syringaresinol > pinoresinol > medioresinol > secoisolariciresinol > lariciresinol > matairesinol and for pineapple, syringaresinol > lariciresinol > matairesinol > secoisolariciresinol > pinoresinol > medioresinol, whereas, the most representative lignan for navel orange was lariciresinol, and secoisolariciresinol for kiwifruit. In berries, as reported by Smeds et al. [78], the most representative lignans among those studied were lariciresinol for cloudberries (5008 µg/100 g dry weight); secoisolariciresinol for blackberries (2902 µg/100 g dry weight), lingoberries (2319 µg/100 g dry weight), blackcurrants (446 µg/100 g dry weight); syringaresinol for cranberries (2578 µg/100 g dry weight), sea buckthorns (1177 µg/100 g dry weight), bilberries (801 µg/100 g dry weight), and red gooseberries (498 µg/100 g dry weight); and pinoresinol for strawberries (1403 µg/100 g dry weight); for raspberries the most representatives were lariciresinol (406 µg/100 g dry weight), syringaresinol (388 µg/100 g dry weight) and pinoresinol (377 µg/100 g dry weight). 

Within the beverage group, a recent work of Angeloni et al. [84] reported, for coffee samples from different Countries, secoisolariciresinol from 27.9 to 52.0 μg L^−1^ and lariciresinol from 5.3 to 27.8 μg L^−1^ respectively, contrary to matairesinol, that was not possible to detect it in each type of coffee.

For foods of animal origin, Kuhnle et al. [72] reported the content of lignans for the first time; in milk and its derived products, the content of dietary lignans was reported (as the sum of secoisolariciresinol, matairesinol, and shonanin) as follows: about 1 µg/100 g wet weight for skimmed, semi-skimmed, or whole milk; in the cheese group, from <1 µg/100 g wet weight for feta cheese derived from ewe’s and goat’s milk, to 4 µg/100 g wet weight for mascarpone, 5 µg/100 g wet weight for parmesan, 6 µg/100 g wet weight for mozzarella (derived from buffalo milk), 13 µg/100 g wet weight for soft Philadelphia cheese (full fat), and to 25 µg/100 g wet weight for Wensleydale cheese. Moreover, cow milk, also condensed and evaporated, showed a content of enterolactone in a range of 3–9 µg/100 g wet weight, and cheese in a range of 3–23 µg/100 g wet weight. 

The same authors [72] reported a dietary lignan content for meat (including different meat cuts and offal) at various cooking of 1–2 µg/100 g wet weight in chicken, 3–9 µg/100 g wet weight in pork, 4–16 µg/100 g wet weight in beef, 4–17 µg/100 g wet weight in lamb; whereas, for eggs, 2–3 µg/100 g wet weight for egg whites and 6–10 µg/100 g wet weight for egg yolks. Small quantities of enterolignans (<6 µg/100 g wet weight) were detected in some type of eggs and meat cuts. 

Most of the foods are consumed after cooking or processing, depending on the type of food matrices and the eating habits of the consumers, indeed, researches are moving in this direction [72,85,86]; indeed, the evaluation of the effects of all type of factors on lignan content in different food matrices increase the reliability of lignan intake estimations.

At the same time, procedures to improve the content of lignans such as milling, parboiling, or supplementation diet in animals [86,87,88] were optimized. 

Nowadays, attention is paid to less common species and agro-industrial side streams [89,90,91], in order to continually explore new sources of lignans. 

## 4. Lignans and Databases: The Current Workflow

Studies that examine the relationship between diet and health have led to increased interest in all biologically active constituents that are present together with nutrients in food, and data on these, as well as other compounds, are increasingly required in the database system.

A complete and comprehensive harmonized databases on the content of lignans in foods are useful in dietary assessment and in the evaluation of formulated diet, in order to be used in observational studies as key elements for healthy nutritional patterns [92]. Knowledge of the dietary intake of lignans is needed for understanding the relationship between a lignan-rich diet and the potential lower risk of development of various diseases, that is, hormone-related cancers, heart diseases, menopausal symptoms, and osteoporosis.

Detailed and accurate information on the lignans in foods is crucial in determining exposure and to investigate health effects in vivo.

To reach this objective, limitations were given by numerous existing factors—from one side, the diversity of the chemical features of compounds, the great number of dietary sources, and the large variability in content from a specified source, to the other side, the different extraction procedures and analytical techniques and methodologies [93]. Additional factors, in some cases, are given by the fact that several studies have been focused only on few compounds within a class, and by the lack of appropriate analytical methods.

In the last decade, researchers are addressing the identification and determination of lignan profiles in main food groups and in food chain products; when a new dataset for nutritional values is used, it is very important to evaluate the quality of the analytical information [55]. New experimental and analytical data on lignan content are now available for updating and expanding food composition databases [64,65,67,68,69,70,71,72,73,74,75,76,77,78,79,80,81,82,83,84]. In Table 1 the main national databases of lignans are described. 

The first examples of databases including lignans were movements toward the development of phytoestrogen databases [67,94]. Valsta et al. [67] reported on expansion of the Finnish National Food Composition Database (Fineli^®^), compiling values for plant lignans, matairesinol, and secoisolariciresinol (shown in Figure 1), and the isoflavones, daidzein and genistein. 

Further, Milder et al. [64] developed a lignan database for 83 solid foods and 26 beverages commonly consumed in the Netherlands: the amount of lignans in plant foods varied widely, from 0 to 301,129 μg/100 g fresh weight; in detail, the lignan values varied from 10 to 30,129 μg/100 g fresh edible weight of oilseeds and nuts, from 7 to 12,474 μg/100 g fresh edible weight of grain products, from 0 to 2321 μg/100 g fresh edible weight of vegetables, from 0 to 450 μg/100 g fresh edible weight of fruits, from 26 to 37 μg/100 g fresh edible weight of legumes, and in beverages ranged from 0 to 91 μg/100 mL. Only five of the studied foods did not contain a measurable amount of lignans and, in most cases, the amount of lariciresinol and pinoresinol was larger than that of secoisolariciresinol and matairesinol.

On the basis of above mentioned lignan databases, in another work, Milder et al. [68] have assessed the lignan intake in a representative sample of 4660 Dutch adults (Dutch Food Consumption Survey, carried out in 1997–1998), reporting the following contribution percentages to lignan intake: lariciresinol and pinoresinol contributed 75%, whereas secoisolariciresinol and matairesinol contributed 25%; and the major food sources of lignans were beverages (37%), followed by vegetables (24%), nuts and seeds (14%), bread (9%), and fruits (7%) [68].

Thompson et al. [69] developed a lignan database of foods consumed in Canada: nine phytoestrogens were identified in 121 food products of Canada by GC–MS, including lignans; decreasing amounts (on wet weight, µg per 100 g) of total lignans are reported in the following order: nuts and oilseeds (25–379012), cereals and breads (2.0–7239.3), legumes (1.8–979.4), fruits (0.3–61.8), vegetables (1.2–583.2), soy products (2.2–269.2), meat products and other processed foods (0.2–415.1), alcoholic beverages (1.1–37.3), and non-alcoholic beverages (0.9–12). Matairesinol was the least-concentrated lignan in most studied foods, whereas secoisolariciresinol reached the highest concentration in 63 foods, lariciresinol in 44 foods, and pinoresinol in 14 foods [69].

Peñalvo et al. [70] have reported the content of six plant lignans (shown in Figure 1) in 86 food items commonly consumed in Japan: the amount of plant lignans ranged from 0 to 1724 μg/100 g (wet basis); in details, as for instance, considering the food group of vegetables, most of the lignans were in the stems and leaves of Japanese parsley, asparagus, Japanese spinach, bitter oranges, and Chinese citrus, and related concentrations in vegetables ranged from 19 to 1724 μg/100 g wet basis.

Moreno-Franco et al. [77] have developed the Aligna databases, by collecting data from scientific publications for alkylresorcinols and lignans in common foods and beverages, and by analyzing foods particularly consumed in Spain; moreover, the assess of lignans intake in Spain was evaluated and reported as follows: 0.76 mg/day, with the major contributors, i.e. oils and fats (33 percent), fruits and vegetables (30 percent), bread (14 percent), and wine and beer (10 percent) [77].

In several works, Kuhnle et al. [71,72,74,75] reported the content of secoisolariciresinol and matairesinol in 115 foods of animal origin, 240 different foods based on fresh and processed fruit and vegetables, 101 cereal and cereal-based foods including bread, breakfast cereals, biscuits, pasta, and rice, and about 40 beverages, nuts, seeds, and oils. The study of Mulligan et al. [81] estimates the average intakes of isoflavones, lignans, enterolignans, and coumestrol in the Norfolk arm of the European Prospective Investigation into Cancer and Nutrition (EPIC-Norfolk) from 7-days food diaries, and provides data on total isoflavone, lignan, and phytoestrogen consumption by food group—the mean daily total lignan intake was 361 (SD 230) µg in soya-consuming men, and 311 (SD 178) µg in non-soya-consuming men; the mean daily total lignans intake was 318 (SD 212) µg in soya-consuming women and 251 (SD 141) µg in non-soya-consuming women [81].

It is worth mentioning the work of Tetens et al. [95] which estimated and evaluated the scale of consumption and the main food sources of lignans in five European countries using the Finnish databases [67], including lignans and Dutch lignan databases [64], respectively; in detail, 42 food groups known to contribute to the total lignan intake were selected and a value attributed for secoisolariciresinol and matairesinol from the Finnish lignan database (Fineli^®^) or for secoisolariciresinol, matairesinol, lariciresinol, and pinoresinol from the Dutch database. The total intake of lignans was estimated from food consumption data for adult men and women (19–79 years) from Denmark, Finland, Italy, Sweden, United Kingdom, and the contribution of aggregated food groups calculated using the Dutch lignan database [75]. The authors showed that, compared to the total lignan intakes among Dutch men and women, the total lignan intakes were higher in Denmark and Sweden, and within similar range in Finland, Italy, and United Kingdom [75].

Here, also, are some examples of utilization of lignan databases to investigate the association between lignan intake and prevention of some chronic pathologies. 

A recent study was undertaken by Witkowska et al. [96] that examined the total and individual lignan intakes and their dietary sources in postmenopausal Polish women: for lignan content, the Dutch lignan database was used [64]; for beverages, nuts, seeds, and oils, data from Kuhnle et al. [71] were taken, and when data on lignan content were missing, values were taken from Thompson et al. [69]; in women with cardiovascular disease (CVD), secoisolariciresinol accounted for 50.15% lignan intake from plant foods, as compared to 44.8% in the control. Pinoresinol, lariciresinol, and matairesinol contributed to the total lignan intakes of CVD and non-CVD women in 24.0% vs. 26.1%, 22.7% vs. 26.1%, and 3.1% vs. 2.9%, respectively [96].

Nowadays, the major core public databases that gather extensive data on the polyphenol content of foods and beverages include lignans—Phenol-Explorer [97], the first comprehensive database on polyphenol content in foods [98] and eBASIS (Bioactive Substances in Food Information Systems) [99,100,101], published through the EuroFIR project.

Phenol-Explorer was the first comprehensive web-based database on polyphenol content in foods and an open-access database and, now, throughout several updates [102,103], includes new data on pharmacokinetic and metabolites, the effect of food processing and cooking and, in the last update (version 3.6), 1451 new content values for lignans have been added (to the database). The development of the Phenol-Explorer database included five main steps: literature search, data compilation, data evaluation, data aggregation, and final data exportation to the MySQL database which is used by the web interface. Composition data were collected from peer-reviewed scientific publications, and evaluated before they were aggregated to produce final representative mean content values. 

The eBASIS database contains composition data and biological effects of over 300 major European plant foods of 24 compound classes, such as glucosinolates, phytosterols, polyphenols, isoflavones, glycoalkaloids, and xanthine alkaloids in 15 EU languages. EuroFIR eBASIS resource is a compilation of expert critically evaluated data extracted from peer-reviewed literature as raw data. This could be seen and considered as the first effort to establish a harmonized food composition information system in EU. Indeed, eBASIS should be defined as the first EU harmonized food composition database. Currently, 2695 data points for lignans were inserted in eBASIS, in detail, 658 values for secoisolariciresinol, 550 values for matairesinol, 313 values for lariciresinol, 276 values for pinoresinol, 93 values for medioresinol, and 86 values for syringaresinol [99,101].

Indeed, considering the importance of metabolic pathways and the benefits of bioactive compounds in humans, it is worth mentioning the Human Metabolome Database or HMDB 4.0 [104], a web metabolomic database on human metabolites including lignans and their metabolites [105], as well as PhytoHub [106], a freely electronic database containing detailed information about all phytochemicals and their metabolites commonly ingested in diets [107].

## Figures and Tables

**Figure 1 molecules-23-03251-f001:**
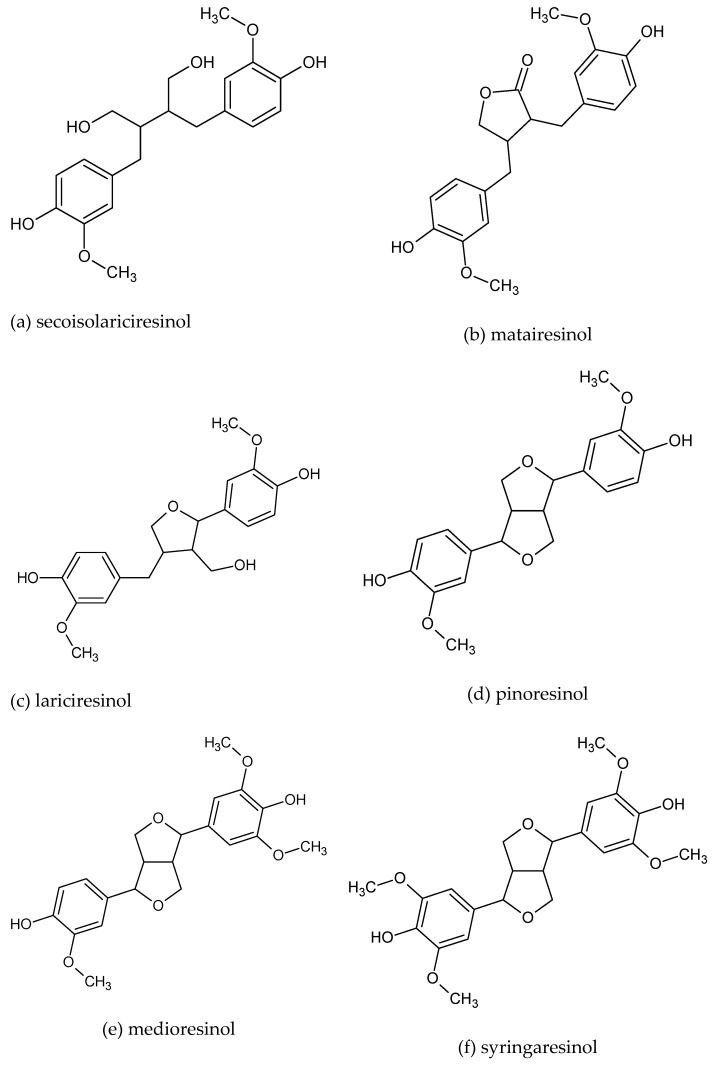
The chemical structure of main dietary lignans, (**a**) secoisolariciresinol, (**b**) matairesinol, (**c**) lariciresinol, (**d**) pinoresinol, (**e**) medioresinol, and (**f**) syringaresinol.

**Figure 2 molecules-23-03251-f002:**
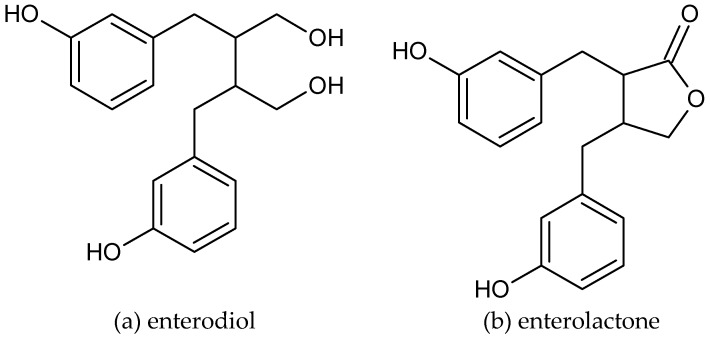
The chemical structure of enterolignans, (**a**) enterodiol and (**b**) enterolactone.

**Table 1 molecules-23-03251-t001:** National databases of lignans.

Country	Type of Database	Main/Common Lignan Compounds	N° Total Foods	Food Groups and Subgroups	References
Finland	Phytoestrogen Database including lignans	Secoisolariciresinol Matairesinol	180	Vegetables, Herbs and spices, Mushrooms, Fruits, Miscellaneous	[67]
Netherland	Lignan Database	Secoisolariciresinol Matairesinol Lariciresinol Pinoresinol	109	Oilseeds and nuts, Grain products, Vegetables and legumes, Fruits, Vegetable oils and fats, Other solid foods, Alcoholic beverages, Non-alcoholic beverages, Juices, Other beverages,	[64]
Canada	Phytoestrogen Database including lignans	Secoisolariciresinol Matairesinol Lariciresinol Pinoresinol	121	Soy products. Legumes. Nuts and oil seeds. Vegetables. Fruits. Cereals and bread. Meat products and other processed foods. Non-alcoholic beverages. Alcoholic beverages	[69]
Japan	Lignan Database	Secoisolariciresinol Matairesinol Lariciresinol Pinoresinol Syringaresinol Medioresinol	86	Vegetables. Tubers and roots. Mushrooms. Fruits. Legumes. Soybean-based products. Cereal-based products. Animal-derived products	[70]
Spain	Alkylresorcinols and Lignans Database	Secoisolariciresinol Matairesinol Lariciresinol Pinoresinol Syringaresinol Medioresinol	593	Vegetables. Grains. Animal. Fats. Drinks	[77]
United Kingdom	Phytoestrogen Database including lignans	Secoisolariciresinol Matairesinol (and Shonanin)	496	Cereal and cereal-based foods, Fresh and processed fruit and vegetables including soya-based foods and legumes, Nuts and seeds, Oils. Alcoholic beverages. Tea and coffee. Dairy products, Eggs, Meat, Fish and seafood	[71,72,74,75]

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
