# Peer review of "Dietary Lignans: Definition, Description and Research Trends in Databases Development"

_molecules, 2018, doi:10.3390/molecules23123251_

Round 1

Reviewer 1 Report

Title:

The title should emphasis that it is a question of databases of dietary lignans or lignans in food.

Abstract:

Lines 13-16: the first sentence is almost incomprehensible.

Introduction:

One of the difficulties of lignan analysis is that there are no standard methods for extraction. This would be something to bring up as well. Also the “new” emerging lignans due to LC combined with HR-MS/MS, have been and will broaden the view regarding dietary lignans (e.g. Hanhineva, K., Rogachev, I., Aura, A.M., Aharoni, A., Poutanen, K., Mykkänen, H. 2012. Identification of novel lignans in the whole grain rye bran by non–targeted LC–MS metabolite profiling. Metabolomics, 8, 399–409).

One good reference describing analytical aspects would be Willför, Smeds & Holmbom. 2006. Chromatographic analysis of lignans.  J. Chromatogr. A. 1112, 64–77. Although after that Smeds et al 2007 [reference 50] and more recently Nørskov & Knudsen 2016 [DOI: 10.1021/acs.jafc.6b03452] have discussed about the analytical aspects quite in-depth.

The Introduction would also benefit the addition of references to some good reviews regarding the health effects of lignans, such as:

Landete 2012. Plant and mammalian lignans: A review of source, intake, metabolism, intestinal bacteria and health.   https://doi.org/10.1016/j.foodres.2011.12.023

Kiyama. 2016. Biological effects induced by estrogenic activity of lignans. https://doi.org/10.1016/j.tifs.2016.06.007

Aehle et al 2011. Lignans as food constituents with estrogen and antiestrogen activity. https://doi.org/10.1016/j.phytochem.2011.08.013

Zamora-Ros et al 2014. Measuring exposure to the polyphenol metabolome in observational epidemiologic studies: current tools and applications and their limits. https://doi.org/10.3945/ajcn.113.077743

Peterson et al 2010. Dietary lignans: physiology and potential for cardiovascular disease risk reduction. https://doi.org/10.1111/j.1753-4887.2010.00319.x

Chapter 3:

-lines 112-121: a sudden jump from food point-of-view to feed. I would suggest carefully to consider removing this part and the references in question because they are not relevant to the lignans & databases.

- line 122: agro-industrial “waste” is in more elegant term a “sidestream”

Chapter “4”:

- line 125: chapter number is missing

- lines 231-233: Sentence need some work.

- line 234: “methabolic” should be “metabolic”

References

there was some inconsistency in the writing style of references

reference 22:  relevancy ?

references 53 and 55 are the same

references 56-58: relevancy ?

Author Response

The authors want to express their gratitude to the reviewer for his/her valuable comments and suggestions. The authors’ replies to the individual points raised are reported in Italic below.

English language and style

( ) Extensive editing of English language and style required 
(x) Moderate English changes required 
( ) English language and style are fine/minor spell check required 
( ) I don't feel qualified to judge about the English language and style 

The linguistic revision of whole manuscript was carried out.

Yes

Can be improved

Must be improved

Not applicable

Does the introduction   provide sufficient background and include all relevant references?

( )

(x)

( )

( )

Is the research design   appropriate?

( )

( )

( )

(x)

Are the methods adequately   described?

( )

( )

( )

(x)

Are the results clearly   presented?

( )

( )

( )

(x)

Are the conclusions   supported by the results?

(x)

( )

( )

( )

Comments and Suggestions for Authors

Title:

The manuscript was improved following your suggestions.

The title should emphasis that it is a question of databases of dietary lignans or lignans in food.

The title was changed in order to emphasis the focus of dietary lignans or lignans in food

Abstract:

Lines 13-16: the first sentence is almost incomprehensible.

 These lines are rewritten to make them clearer.

Introduction:

One of the difficulties of lignan analysis is that there are no standard methods for extraction. This would be something to bring up as well.

As you suggested lines on extraction methodologies and importance of extraction procedures was inserted and related references.

Also the “new” emerging lignans due to LC combined with HR-MS/MS, have been and will broaden the view regarding dietary lignans (e.g. Hanhineva, K., Rogachev, I., Aura, A.M., Aharoni, A., Poutanen, K., Mykkänen, H. 2012. Identification of novel lignans in the whole grain rye bran by non–targeted LC–MS metabolite profiling. Metabolomics, 8, 399–409).

This proper reference was inserted.

One good reference describing analytical aspects would be Willför, Smeds & Holmbom. 2006. Chromatographic analysis of lignans.  J. Chromatogr. A. 1112, 64–77. Although after that Smeds et al 2007 [reference 50] and more recently Nørskov & Knudsen 2016 [DOI: 10.1021/acs.jafc.6b03452] have discussed about the analytical aspects quite in-depth.

These references that well describe the analytical aspects were inserted.

The Introduction would also benefit the addition of references to some good reviews regarding the health effects of lignans, such as:

Landete 2012. Plant and mammalian lignans: A review of source, intake, metabolism, intestinal bacteria and health.   https://doi.org/10.1016/j.foodres.2011.12.023

Kiyama. 2016. Biological effects induced by estrogenic activity of lignans.https://doi.org/10.1016/j.tifs.2016.06.007

Aehle et al 2011. Lignans as food constituents with estrogen and antiestrogen activity.https://doi.org/10.1016/j.phytochem.2011.08.013

Zamora-Ros et al 2014. Measuring exposure to the polyphenol metabolome in observational epidemiologic studies: current tools and applications and their limits. https://doi.org/10.3945/ajcn.113.077743

Peterson et al 2010. Dietary lignans: physiology and potential for cardiovascular disease risk reduction. https://doi.org/10.1111/j.1753-4887.2010.00319.x

Proper references that you suggested are inserted together with  additional lines of description of the health effects of lignans.

Chapter 3:

-lines 112-121: a sudden jump from food point-of-view to feed. I would suggest carefully to consider removing this part and the references in question because they are not relevant to the lignans & databases.

As  you suggested the lines 112-121 and related reference were deleted. Only lines 111-112 and related references  are maintained in the text.

- line 122: agro-industrial “waste” is in more elegant term a “sidestream”

 As you suggested “waste” was replaced by “sidestream”.

Chapter “4”:

- line 125: chapter number is missing

Chapter number was inserted.

- lines 231-233: Sentence need some work.

This sentence was modified to make it clearer.

- line 234: “methabolic” should be “metabolic”

 “methabolic” was corrected into “metabolic”.

References

there was some inconsistency in the writing style of references

reference 22:  relevancy ?

reference 22 was deleted.

references 53 and 55 are the same

the duplicate of reference  was deleted.

references 56-58: relevancy ?

As reported above, 57-58 were deleted, whereas 56 was maintained as one of examples of procedures to increase the content of  lignans  in food.

Reviewer 2 Report

This perspective by Durazzo et al looks to communicate the current status of lignans and their associated databases, particularly in the context of food. I find this perspective to be interesting and would be of interest to readers of this special issue - for this reason, I agree to publication of this work, provided the following changes were made: 

I find that the abstract is very difficult to follow and strongly suggest that it is completely rewritten to better and more clearly communicate the purpose of the perspective. 

In the introduction, the authors refer to some examples of lignan compounds. I would recomment giving the structure of these names compounds. Furthermore, I would suggest that the authors briefly describe the general structures of lignans (i.e. they are formed from the oxidative dimerisation of 2 or more phenyl propanoid units) and name some of the commonly-found types of lignan (perhaps this can be combined with some of the named examples of naturally-occurring lignans). 

There are a number of english-language errors throughout the article - these should be corrected. 

In Section 3 with regard to the occurrence of lignans in food - overall statements are made, but not specific examples are given - I would like further description of the type of lignans (maybe the most commonly found in a few of the matrices given) and a range of their abundance/concentration in these natural sources - what amounts are common? 

In the "Lignans and Databases: current workflow" section, I would, again, recommend the inclusion of some lignan structures - at a minimum the four lignans repeatedly referred to in this section. I would also suggest a table of some sort, summarizing the information in this section, to be added. I would also like additional and more extensive quoting of amounts of the lignans, in this section. 

Author Response

The authors want to express their gratitude to the reviewer for his/her valuable comments and suggestions. The authors’ replies to the individual points raised are reported in Italic below.

English language and style

(x) Extensive editing of English language and style required 
( ) Moderate English changes required 
( ) English language and style are fine/minor spell check required 
( ) I don't feel qualified to judge about the English language and style 

The linguistic revision of whole manuscript was carried out.

Yes

Can be improved

Must be improved

Not applicable

Does the   introduction provide sufficient background and include all relevant   references?

( )

( )

(x)

( )

Is the research   design appropriate?

( )

( )

( )

(x)

Are the methods   adequately described?

( )

( )

( )

(x)

Are the results   clearly presented?

( )

( )

( )

(x)

Are the   conclusions supported by the results?

( )

( )

( )

(x)

The whole manuscript was revised following the suggestions.

Comments and Suggestions for Authors

This perspective by Durazzo et al looks to communicate the current status of lignans and their associated databases, particularly in the context of food. I find this perspective to be interesting and would be of interest to readers of this special issue - for this reason, I agree to publication of this work, provided the following changes were made: 

Thank you so much for your comment and suggestions.

I find that the abstract is very difficult to follow and strongly suggest that it is completely rewritten to better and more clearly communicate the purpose of the perspective. 

As you suggested the abstract was completely  rewritten.

In the introduction, the authors refer to some examples of lignan compounds. I would recomment giving the structure of these names compounds. Furthermore, I would suggest that the authors briefly describe the general structures of lignans (i.e. they are formed from the oxidative dimerisation of 2 or more phenyl propanoid units) and name some of the commonly-found types of lignan (perhaps this can be combined with some of the named examples of naturally-occurring lignans). 

As you suggested, the structure backbone of lignans was described. The chemical structure of main common dietary lignans and enterolignans are inserted.

There are a number of english-language errors throughout the article - these should be corrected. 

The English language errors throughout the article were checked and corrected.

In Section 3 with regard to the occurrence of lignans in food - overall statements are made, but not specific examples are given - I would like further description of the type of lignans (maybe the most commonly found in a few of the matrices given) and a range of their abundance/concentration in these natural sources - what amounts are common? 

Following your suggestions, some examples for the main dietary lignans and their occurrence in food group with the ranges  of concentrations is reported, from researches in literature applying different methodological approaches.

In the "Lignans and Databases: current workflow" section, I would, again, recommend the inclusion of some lignan structures - at a minimum the four lignans repeatedly referred to in this section. I would also suggest a table of some sort, summarizing the information in this section, to be added. I would also like additional and more extensive quoting of amounts of the lignans, in this section. 

Figure 1 with the chemical structure of dietary lignans present in National Databases are reported in previous section and in the text of section 4 was mentioned and referred to Figure. A Table summarizing the main National Databases of lignans are added and additional information of amount of  lignans for each food group are inserted in the text.

Round 2

Reviewer 2 Report

The authors have taken on board all of my recommendations and suggested changes. I can see that they have extensively edited the language used, rewritten the abstract and included the extra figures and tables that I requested. I am supportive of the manuscript being published in its present form.